# Epigenetic Mechanisms in Hirschsprung Disease

**DOI:** 10.3390/ijms20133123

**Published:** 2019-06-26

**Authors:** Ana Torroglosa, Leticia Villalba-Benito, Berta Luzón-Toro, Raquel María Fernández, Guillermo Antiñolo, Salud Borrego

**Affiliations:** 1Department of Maternofetal Medicine, Genetics and Reproduction, Institute of Biomedicine of Seville (IBIS), University Hospital Virgen del Rocío/CSIC/University of Seville, 41013 Seville, Spain; atorroglosa-ibis@us.es (A.T.); leticia.villalba.benito@hotmail.es (L.V.-B.); berta.luzon@ciberer.es (B.L.-T.); raquelm.fernandez.sspa@juntadeandalucia.es (R.M.F.); guillermo.antinolo.sspa@juntadeandalucia.es (G.A.); 2Centre for Biomedical Network Research on Rare Diseases (CIBERER), 41013 Seville, Spain

**Keywords:** enteric nervous system development, Hirschsprung disease, neural crest cells, epigenetic mechanisms

## Abstract

Hirschsprung disease (HSCR, OMIM 142623) is due to a failure of enteric precursor cells derived from neural crest (EPCs) to proliferate, migrate, survive or differentiate during Enteric Nervous System (ENS) formation. This is a complex process which requires a strict regulation that results in an ENS specific gene expression pattern. Alterations at this level lead to the onset of neurocristopathies such as HSCR. Gene expression is regulated by different mechanisms, such as DNA modifications (at the epigenetic level), transcriptional mechanisms (transcription factors, silencers, enhancers and repressors), postranscriptional mechanisms (3′UTR and ncRNA) and regulation of translation. All these mechanisms are finally implicated in cell signaling to determine the migration, proliferation, differentiation and survival processes for correct ENS development. In this review, we have performed an overview on the role of epigenetic mechanisms at transcriptional and posttranscriptional levels on these cellular events in neural crest cells (NCCs), ENS development, as well as in HSCR.

## 1. Introduction

Hirschsprung disease (HSCR, OMIM 142623) is a rare congenital disorder that occurs in approximately one per 5000 live births. It is characterized by the absence of enteric ganglia in the rectum and a variable continuous segment of the proximal intestine resulting in intestinal dysfunction. Based on the length of the aganglionic segment, the pathology is classified as short segment HSCR, long segment HSCR, and total colonic aganglionosis or total intestinal aganglionosis. Most often the disease appears as sporadic HSCR, although familial cases have also been reported. Although 70% of HSCR cases appear without any additional clinical manifestations (isolated HSCR), the remaining 30% of cases manifest with other disorders or congenital malformations (syndromic HSCR) [1].

HSCR results from a failure to fully colonize the gut by enteric precursor cells (EPCs) derived from neural crest cells (NCCs). Such incomplete gut colonization is due to alterations in EPCs proliferation, survival, migration and differentiation during enteric nervous system (ENS) development [2]. Various pathways have been described in relation with these cellular events. Among them, the main pathways described are RET/GFRα1/GDNF and EDNRB/EDN3/ECE1, as well as several transcription factors (PAX3, SOX10, ZFHX1B and PHOX2B) and some morphogens (netrins, semaphorins and SHH) [3]. Therefore, ENS formation is a complex process in which a large number of molecules that are tightly regulated by a specific gene expression pattern are implicated. In this sense, the epigenetic mechanisms among others are involved in the regulation of gene expression, being such regulatory processes an emerging research area in the field of ENS development and specifically in HSCR.

Epigenetic events are defined as “the structural adaptation of chromosomal regions so as to register, signal or perpetuate altered activity state” [4]. Mostly, modifications at this level are stable and are transmitted along generations, but the effect of the environmental agents has also been described [5]. Here, we emphasize the impact of alterations in epigenetic mechanisms such as methylation of DNA, post-translational modifications to histone proteins, polycomb repression, ATP dependent chromatin remodeling and non-coding RNA (ncRNA) [6,7,8,9].

In addition, in this review, we also highlight the above-mentioned regulatory mechanisms that may finally result in the onset of HSCR.

## 2. DNA Methylation

DNA methylation involves the insertion of a CH3 group at the C-5 position of the cytosine ring of DNA. Susceptibility regions of methylation (CpG islands) represent around 40% of all mammalian promoters. These regions are usually unmethylated when gene expression occurs [10]. DNA methylation is essential for many biological processes during mammal development [11,12]. This mechanism is carried out by the enzyme family known as DNA methyltransferases (DNMTs): DNMT1, DNMT3 (A and B). DNMT1 is the maintenance methyltransferase whereas both DNMT3s are *de novo* methyltransferases [13,14].

With respect to *DNMT3A* and its paralog *DNMT3B*, some evidence has shown their crucial role for normal mammal development, as well as their involvement in diseases [15,16,17,18]. Specifically, *Dnmt3A* homozygous knockout mice die some weeks after birth, and rostral neural tube defects and growth impairment have been observed for *Dntm3B* homozygous knockout embryos that finally leads to death, suggesting that both enzymes are essential during embryonic development [19]. Various studies have showed the potential involvement of both genes in NCC development. Specifically, in neural crest cells of chicken embryos, *Dnmt3A* downregulation leads to a reduced expression of genes which is directly implicated in neural crest specification (*Snail, FoxD3, Sox10, Pax7* and *Pax3*) [20]. On the contrary, *Dnmt3B* is upregulated during chicken embryo neural crest formation [21]. Regarding studies in humans, *DNMT3B* mutations have been found in the immunodeficiency-centromeric instability-facial anomalies syndrome (OMIM#242860) [15,22]. In embryonic stem cells, *DNMT3B* knockdown leads to early neural crest differentiation as well as the upregulation of neural crest specifier genes [23]. The contribution of DNMT3B to the onset of HSCR was demonstrated because its downregulation in EPCs from HSCR patients *versus* controls correlated with a decrease of global DNA methylation levels. In addition, the synergistic effect of mutations in both *DNMT3B* and other HSCR–related genes on the severity of the phenotype in HSCR patients has been reported [24]. Such alterations resulted in an altered gene expression pattern [25] and an arrest of cell cycle of the EPCs through P53-P21 activity [26]. Therefore, all this evidence suggests the involvement of *DNMT3B* as a susceptibility gene for HSCR and demonstrates the crucial role of DNA methylation in ENS development and in the onset of HSCR.

Aberrant DNA methylation patterns affecting genes related to ENS development and HSCR have been described. The *RET* proto-oncogene encodes a receptor tyrosine kinase that plays crucial roles in ENS development. It is the main gene associated with HSCR, contains a 5′-CG-3′ rich region within its promoter, and the methylation levels of this region have been demonstrated to be related to its expression level in peripheral white blood cells from HSCR patients [27]. *GFRA4* has been widely proposed as a susceptibility gene in the pathogenesis of HSCR [28]*.* It encodes for a RET co-receptor inducing neuronal survival and differentiation [29] through its interaction with members of the glial derived neurotrophic factors family [30] since RET-GDNF is one of the main pathways related to HSCR, as mentioned above. It has been described that the downregulation of *GFRA4* in HSCR can be partly due to hypermethylation at its promoter region. Therefore, it has been proposed that DNA methylation contributes to the regulation of the neuroprotective role of *GFRA4* on NCCs [31]. *EDNRB* (endothelin receptor type B) is another susceptibility gene for HSCR because the Endothelin 3-Endothelin Receptor B Signalling Pathway is crucial for the correct formation of enteric ganglia [32]. Tang et al. demonstrated that epigenetic inactivation of *EDNRB* might play a role in ENS development and in the onset of HSCR. Specifically, the upregulated expression level of *EDNRB* in HSCR tissue compared with controls correlated with a significantly lower ratio of its methylation level in these patients. [33]. Additionally, it has been described that methylation levels of the sonic hedgehog (*SHH*) promoter are significantly increased in patients with congenital anorectal malformations (ARM), which is correlated with lower levels of its expression [34]. This epigenetic modification on *SHH* may be responsible for abnormal ENS development, which is related to the onset of ARM. De Pontual et al. have identified an aberrant CpG dinucleotide methylation within the *PHOX2B* promoter in neuroblastoma, an embryonic tumor originating from NCCs. This outcome suggests that aberrant methylation patterns within *PHOX2B* might be also implicated in this pathology [35].

Furthermore, an important role of the methylation level of genes that encode for microRNA (miRNA) has also been described in HSCR. In this sense, miR-141, which belongs the tomiR-200 family that has been highly associated with different human pathologies [36], showed that hypermethylation of a CpG Island within its promoter correlated with its downregulation and the subsequent upregulation of its target genes (*CD47* and *CUL3*) in colon tissues from HSCR patients compared with controls. Moreover, such upregulation of CD47 and CUL3 reduced proliferation and migration of 293T (sub-line of adenovirus-immortalized human embryonic kidney cells) and SH-SY5Y (subline of the neuroblastoma cell line SK-N-SH) cell lines. These results suggest that the methylation status of the promoter of the *miR-141* gene might be a key factor in the pathogenesis of HSCR [37].

## 3. Histone Modifications

Histones are the main binding proteins associated with chromatin, and their association with the compacted DNA strand results in nucleosomes. Each nucleosome consists of four duplicated units of histones (H2A, H2B, H3 and H4), resulting in a structure formed by the combination of eight histones (nucleosome core) around which DNA rolls up with unstructured tails [38]. There are several posttranslational modifications described for the evolutionarily conserved histone tails (methylation, acetylation, deacetylation, phosphorylation, ubiquitination, and/or sumoylation) that regulate gene expression [39,40].

Specifically in eukaryotic cells, histone acetylation is established by two different enzymes, histone acetyltransferases (HATs) and histone deacetylases (HDACs) [41,42,43]. Histone acetylations closely associated with open chromatin are related to gene expression (i.e., H3K27ac) [44,45,46,47,48], whereas histone deacetylation is used to close chromatin, which conducts the repression of gene expression [49].

Various histone acetylation and methylation mechanisms have been associated with NCC development, although their implication in HSCR is still unknown. All this evidence suggests a potential role in the onset of this pathology that should be investigated.

In most cases, the histone acetylation corresponds to cis-regulatory regions in the neural crest. The HDAC repression complex promotes trunk crest cell specification [50] and regulates the migration of NCC [51,52,53]. In addition, HDACs take part in regulating downstream NCCs differentiation. In zebrafish, hdac1 and hdac4 are implicated in various developmental events and are expressed during neural crest cell differentiation [54,55,56]. Regarding human development, HDAC4 also relates to syndromes and other diseases derived from neural crest development [57,58]. For instance, brachydactyly mental retardation syndrome has been associated with haploinsufficiency of HDAC4, including craniofacial and skeletal abnormalities (OMIM#600430) [59]. HDAC3 and HDAC8 are related to the regulation of smooth muscle cell differentiation and cardiac outflow tract development in mice [60]. Specifically, HDAC8 epigenetically regulates skull morphogenesis in NCCs by inhibiting Lhx1 and Otx2 activity [61]. HDACs can maintain or induce the active gene state [62] together with HATs. In this sense, it has been shown that the binding of specific HDACs (HDAC1 and HDAC2) to promoters causes NCC differentiation to peripheral glia [63]. In zebrafish, *hdac1* is required for eye development, the central nervous system and NCC populations [54,64,65,66,67]. Ignatius et al. showed the specific requirements for *hdac1* function during the development of the neural crest in zebrafish and therefore in ENS formation [54].

Regarding histone methylation, JMJD2A mediates neural crest gene expression by modulating the epigenetic modification H3K9m3, that finally establishes the neural crest in the embryonic stage. *JmjD2A* knockdown resulted in a drastic loss of Snail2, FoxD3 and Sox10 expression. When H3K9m3 modification is present in the promoter regions of Sox10 and Snail2, their interaction with JMJD2A is unraveled by Chromatin immunoprecipitation (ChIP) assays [68]. Thus, *JmjD2A* is necessary to correct neural crest establishment during embryo development. Moreover, PHF8, can demethylate the H4K20me1 and H3K9me1 marks close to the start of the transcription site that turns active. Interestingly, this transcriptional regulator was previously associated with the regulation of neural crest development in diverse vertebrate models [69,70,71,72].

The only histone modulator factor that has been related to HSCR thus far is *MECP2* (Methyl-CpG binding Protein 2). Its association with HDACs and histone methyltransferases (HTMs) forms stable repressor complexes for gene expression [73]. Zhou et al. identified a decrease in the expression levels of *MECP2* in HSCR patients and, interestingly, the downregulation of this gene in SH-SY5Y caused a decline in cell proliferation. Nevertheless, in the methylation level of MECP2, there was no difference when analyzing both groups. Moreover, similar outcomes were found in miR-34b, which is a regulator of *MECP2* expression. These results suggest that alteration in the expression level of MECP2 may be relevant in the etiology of HSCR through the regulation of histone modifications [74].

## 4. Polycomb Repressive Complex (PRC)

This complex is formed by a series of proteins that prevent the transcription of their target genes by catalyzing H3K27me3 epigenetic complex [75,76]. There are two classes of PRC, PRC1 and PRC2, which are implicated in embryonic development and differentiation of neural crest-derived craniofacial structures [77,78]. In this sense, the differentiation of the cranial neural crest in chondrocytes has been reported to be established by EZH2 (the enhancer of the zeste homolog 2), which is a subunit of PRC2, as well as Ring1b/Rnf2 (the single E3 ubiquitin ligase) in PRC1 [77,78,79].

Heterozygous mutant mice, with respect to the *Aebp2* gene, which encodes for a component of PRC2 expressed in NCCs [80], show similar phenotypes to HSCR and Waardenburg syndrome patients. Both pathologies arise from defects in the development of NCCs [81,82,83]. Interestingly, these mutants showed an alteration in the neural crest gene expression levels, such as the lower expression of *Sox10*. This result is similar to the reduced SOX10 dosage frequently observed in Waardenburg syndrome type 4 [83]. Therefore, *Aebp2* misregulation might be responsible for HSCR and the Waardenburg syndrome due to an aberrant epigenetic regulation of neural crest genes. In the same way, *EED* (Embryonic Ectoderm Development), one of the two core catalytic subunits of PRC2 has been described as a regulator of neural crest gene expression during NCC determination and migration [84]. In the HSCR context, there is a significant upregulation of *EED* in EPCs from HSCR patients with respect to controls [25].

## 5. ATP-Dependent Chromatin Remodeling

This epigenetic mechanism is mediated by protein complexes, such as CHD (chromodomain helicase DNA-binding), ISWI (imitation switch) and SWI/SNF (mating-type switch/sucrose nonfermenting), that change the structure of chromatin by ATP-hydrolysis. They promote regions with a lack of nucleosomes to facilitate transcription factor binding and binding of other regulatory proteins in these regions [85,86]. Specifically, CHD7 (Chromodomain Helicase DNA Binding Protein 7) together with PBAF (SWI/SNF) [87] induce neural crest specification in embryonic stem cells from humans [88]. Williams syndrome transcription factor is a subunit of WICH and WINAC, both being ATP-dependent chromatin remodeling complexes [89]. Such genes are transcription factor is related to William’s syndrome (OMIM#194050), a developmental disorder that shows alterations in neural crest-derived tissues [90,91]. In summary, all this evidence suggests a possible involvement of this mechanism in ENS formation and therefore in HSCR.

## 6. NcRNA

Several classes of ncRNA have emerged to play key roles in modulating many cellular processes, such as micro RNA (miRNAs), long non-coding RNAs (lncRNAs) and circular RNAs (circRNAs). In particular, miRNAs are highly conserved RNAs (20–24 nucleotides) that inhibit gene expression by posttranscriptional mechanisms through complementary binding to the 3′-untranslated regions (3′-UTR) of target mRNA [92,93]. LncRNAs are defined as RNA transcripts longer than 200 bp which do not encode for proteins. They play an important regulatory role in gene expression through epigenetic mechanisms (chromatin remodeling, transcriptional and posttranscriptional processing) that finally will determine diverse cellular processes [94]. Finally, circRNAs form covalently closed continuous loop structures through specific splicing methods and work as transcription regulators [95] or as miRNA sponges [96].

Different research studies have related miRNAs, lncRNA and circRNA to HSCR [28,37,97,98,99,100,101,102,103,104,105,106,107,108,109,110,111,112,113,114,115,116,117,118,119,120,121]. These associations have been based on either their differential expression in HSCR tissues, an aberrant expression of their target genes and, finally, alterations in migration, proliferation and/or apoptosis processes of NCCs during development (Table 1). Their potential role in cellular processes has been analyzed through in vitro approaches using various cell lines such as 293T and SH-SY5Y. Therefore, although much evidence suggests a relationship between ncRNA and HSCR, the role of these ncRNAs in the onset of the disease should be thoroughly clarified.

Finally, a relationship between 3′UTR *RET* variants and HSCR has been widely described. Fitze et al. characterized several *RET* polymorphisms in a group of HSCR patients and controls and found two variants located at the 3′UTR, c.3187+47T>C (rs2075912) and 3′UTR+124A>G) with a strong association with HSCR [122]. In contrast, Griseri et al. identified a “protective” *RET* haplotype characterized by the presence of an SNP, g.128496T>C (rs3026785) in the 3′UTR of *RET* [123]. Moreover, they suggested that the protective effect against HSCR of this allele might be due to lower mRNA degradation, which leads to an increase of gene transcripts and probably an increase in the amount of total RET protein. Similarly, Pan et al. screened the *RET* 3′UTR in the Chinese population and identified a combination of 7 SNPs that seems to act as protective haplotypes [124]. Implication in the miRNA-mediated regulation of gene expression of these *RET* polymorphisms still needs to be further elucidated.

## 7. Conclusions

HSCR is a human congenital disorder due to an incorrect process in ENS formation attributed to an aberrant migration, proliferation, differentiation or survival of NCCs. Several epigenetic events have been related to ENS development and HSCR. Nevertheless, their potential role in the context of HSCR is just beginning to be defined. In this review, we have summarized the epigenetic mechanisms at transcriptional and posttranscriptional levels implicated in NCCs, ENS development and HSCR described so far (Figure 1). Nonetheless, additional studies are needed to improve the knowledge about the role of epigenetics in the pathogenesis of HSCR.

## Figures and Tables

**Figure 1 ijms-20-03123-f001:**
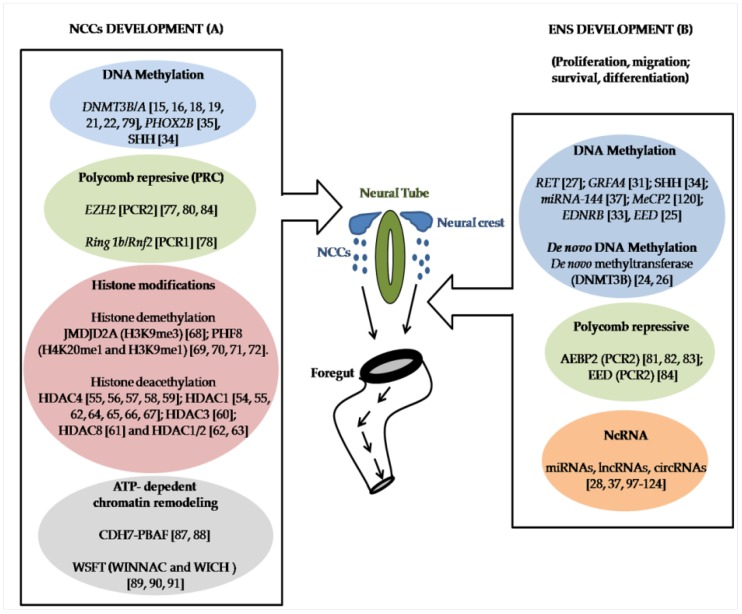
Scheme of the epigenetic processes implicated in Neural Crest Cells (NCCs) and Enteric Nervous System (ENS) development in HSCR context. Epigenetic mechanisms identified in NCC development (**A**) and in ENS formation (**B**) with a possible role in the onset of the disease. Adapted from http://www.columbia.edu/itc/hs/medical/humandev/2006/HD10/ENS6.pdf.

**Table 1 ijms-20-03123-t001:** NcRNAs with a potential role into the pathogenesis of HSCR.

Role on Cellular Processes	ncRNA/Reference	Expression in HSCR Tissue	Change
proliferation and migration	miR-141 [37]	downregulated	↑CD47/CUL3
	miR-195 [104]	upregulated	↓DIEXF
	miR-200a/141 [106]	downregulated	↑PTEN
	miR-206 [102,112]	downregulated	↑SDPR/FN1
	miR-192/215 [121]	downregulated	↑NID1
	miR-218-1 [115]	upregulated	↑SLIT2 ↓RET/PLAG1
	miR-215 [103]	downregulated	↓IARS2/↑SIGLEC-8
	miR-369-3p [110]	upregulated	↓SOX4
	miR-483-3p [119]	downregulated	↓IGF2 ↑FHL1
	miR-214 [117]	upregulated	↓PLAGL2
	HOTTIP [118]	downregulated	↓HOXA13
	miR143HG [100]	upregulated	↓miR-143/↑RBM24
	AFAP1-AS [99]	downregulated	↑miR-181a/↓RAP1B
	MEG3 [105]	downregulated	↓miR-770-5p/↑SRGAP1
	FAL1 [107]	downregulated	↓AKT1
	miR31HG [97]	downregulated	↓miR-31/31*
	LOC100507600 [114]	downregulated	↑miR128–1-3p/↓BMI1
	cir-ZNF609 [111]	downregulated	↑miR-150-5p/↓AKT3
	circ-PRKCI [120]	downregulated	↑miR-1324/↓PLCB1
	cir-CCDC66 [116]	downregulated	↑miR-488-3p/↓DCX
proliferation and apoptosis	miR-483-5p [28]	upregulated	↓GFRA4
proliferation	miR-939 [98]	upregulated	↓LRSAM1
	LOC101926975 [113]	downregulated	↓FGF1
apoptosis	HN12 [101]	upregulated	-
Unknown	HA117 [108,109]	upregulated	↓DPF3/FOXA1/DUSP6

↑: upregulation/↓: downregulation.

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
