# Peer review of "Epigenetic Mechanisms in Hirschsprung Disease"

_ijms, 2019, doi:10.3390/ijms20133123_

Round 1
Reviewer 1 Report
In the current review article titled ‘Epigenetic mechanisms in Hirschsprung disease’, the authors discussed the role of epigenetic mechanisms in the field of enteric nervous system (ENS) development and specifically in Hirschsprung disease (HSCR). The authors emphasize the impact of alterations in epigenetic mechanisms such as DNA methylation, histone modifications, polycomb repression, ATP dependent chromatin remodeling and non-coding RNA.
Over all after following minor corrections, manuscript is good for publication.
Page # 2: Lines # 27: There are two periods.
development. .
Page # 2: Lines # 28: There are multiple extra commas.
(, Snail, ,
Page # 3: Lines # 22:
More detailed information of the cell lines, 293T and SH-SY5Y are needed. For example, derivation, species, normal cell or immortalized cell?
Page # 3: Lines # 43:
HDCA4 may be HDAC4.
Page # 3: Lines # 46:
HDCA8 may be HDAC8.
Page # 4: Lines # 2 & # 3: The font style is different between “hdac1”.
Page # 4: Lines # 7: There are two periods.
Sox10 expression. .
Page # 5: Lines # 4: There is an unknown word.
HSCR.6. NcRNA
Figure 1:
The schematic figure is too complicated to understand. The letters are too small to read. I think that the information of gene expression is not necessary in this figure because it is not definite. Readers can get the information from the references cited in the figure. More concise and comprehensive scheme will be needed.
Author Response
Point 1. Over all after following minor corrections, manuscript is good for publication.
Page # 2: Lines # 27: There are two periods.
development. .
Page # 2: Lines # 28: There are multiple extra commas.
(, Snail, ,
Page # 3: Lines # 22:
More detailed information of the cell lines, 293T and SH-SY5Y are needed. For example, derivation, species, normal cell or immortalized cell?
Page # 3: Lines # 43:
HDCA4 may be HDAC4.
Page # 3: Lines # 46:
HDCA8 may be HDAC8.
Page # 4: Lines # 2 & # 3: The font style is different between “hdac1”.
Page # 4: Lines # 7: There are two periods.
Sox10 expression. .
Page # 5: Lines # 4: There is an unknown word.
HSCR.6. NcRNA
Response 1: We apologize for these mistakes. We have amended all the minor points indicated by the reviewer, We apologize for these mistakes. We have amended all the minor points indicated by the reviewer, they are highlighted in blue.
Point 2. Figure 1:
The schematic figure is too complicated to understand. The letters are too small to read. I think that the information of gene expression is not necessary in this figure because it is not definite. Readers can get the information from the references cited in the figure. More concise and comprehensive scheme will be needed.
Response 2: We appreciate the suggestions made by the reviewer about the schematic figure.In this sense we have performed all the changes required to clarify the figure.
Reviewer 2 Report
Hirschsprung disease is known to be a rare congenital disease very difficult to be curable.
So it is very important to understand the mechanism associated with the development of this disease in genetic background.
This paper describes about this topic and the contents appears to be sound.
Though Hirschsprung disease is a rare congenital disease, the underlying mechanism might be associated with the appearance of similar clinical findings such as constipation in geriatric medicine.
So, the readers might be interested for this topic.
In my opinion, it will be good if this comments can be reflected for the description of introduction.
Author Response
Point 1: Hirschsprung disease is known to be a rare congenital disease very difficult to be curable.
So it is very important to understand the mechanism associated with the development of this disease in genetic background.
This paper describes about this topic and the contents appears to be sound.
Though Hirschsprung disease is a rare congenital disease, the underlying mechanism might be associated with the appearance of similar clinical findings such as constipation in geriatric medicine.
So, the readers might be interested for this topic.
In my opinion, it will be good if this comments can be reflected for the description of introduction.
Response 1: We appreciate the opinion of the reviewer and agree with these comments about the crucial role of the genetic background in HSCR. Extensive research over the past few decades has provided important insights that contribute in the onset of HSCR, involving either genetic, epigenetic and environmental factors. In the current manuscript we have focused exclusively on epigenetic mechanisms associated with the disease.